# Embracing Evolution: A Call for Body-Control Co-Design in Embodied Humanoid Robot

## Abstract

Humanoid robots, as general-purpose physical agents, must integrate both intelligent control and adaptive morphology to operate effectively in diverse real-world environments. While recent research has focused primarily on optimizing control policies for fixed robot structures, this position paper argues for *"evolving both control strategies and humanoid robots' physical structure under a co-design mechanism"*. Inspired by biological evolution, this approach enables robots to iteratively adapt both their form and behavior to optimize performance within task-specific and resource-constrained contexts. Despite its promise, co-design in humanoid robotics remains a relatively underexplored domain, raising fundamental questions about its feasibility and necessity in achieving true embodied intelligence. To address these challenges, we propose practical co-design methodologies grounded in strategic exploration, Sim2Real transfer, and meta-policy learning. We further argue for the essential role of co-design by analyzing it from methodological, application-driven, and community-oriented perspectives. Striving for guiding and inspiring future studies, we present open research questions, spanning from short-term innovations to long-term goals. This work positions co-design as a cornerstone for developing the next generation of intelligent and adaptable humanoid agents.

## 1 Introduction

As an emerging research area, Embodied AI posits that intelligence stems from an agent's ability to actively explore, interact with, and learn from its environment in a continuous and dynamic manner. Within this learning paradigm, recent studies have developed various robot control models based on deep neural network backbones, enabling scalability across diverse tasks and environments [1, 2, 3, 4].

In studies of embodied robot agents, their skills are closely tied to the physical form. For example, robot arms, grippers, and dexterous hands are commonly employed for manipulation tasks such as grasping, placing, and assembling objects [5, 6]. Similarly, wheeled robots, bipedal robots, and quadrupedal robots are designed for locomotion tasks, including walking, climbing, and navigation [7]. To develop general-purpose robots, recent studies have focused on humanoid robots. Equipped with dual arms, legged body, and advanced sensors, humanoid robots are well-suited for a wide range of mobile locomotion tasks, enabling them to seamlessly handle everyday tasks [8, 9, 10, 11].

In recent years, the development of humanoid robots has primarily centered on control policy design, typically built upon predefined physical structures. These robotic designs are often the result of manual engineering and domain-specific heuristics, which are rarely subject to optimization within the embodied humanoid system. However, embodied intelligence is not solely determined by control performance, but is also fundamentally grounded in agents' physical structure [12]. For instance, in natural systems, organisms evolve their body morphology to adapt to changing environmental conditions. Similarly, embodied agents should incorporate evolutionary mechanisms to adapt to task requirements and environmental dynamics.

An effective method of realizing such evolutionary mechanisms is the robotic co-design problem, which seeks to jointly optimize both the control policy and the morphological design of robotic

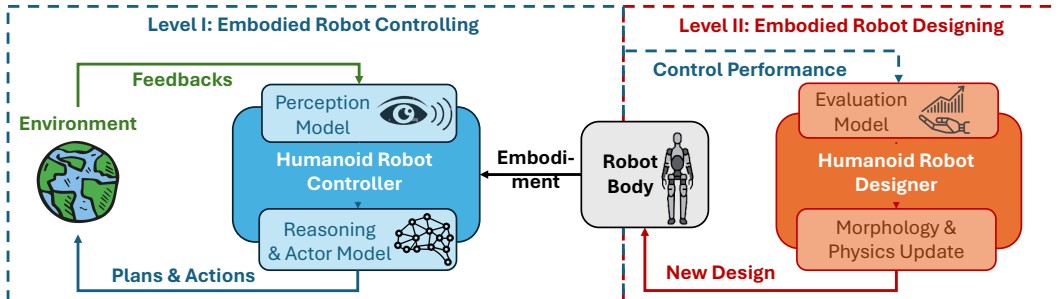

Figure 1: The co-design framework for humanoid robots, which can be formulated as a bi-level optimization problem, consisting of two interconnected phases: 1) learning the control policy for the humanoid robot, and 2) designing the robot's physical structure (Section 3.1).

systems [13]. While prior studies have explored co-design in quadruped robots [14, 15], soft robots [16], bi-piedal robots [17, 18] and modular robots [19, 20] (see Table 1), its extension to more advanced humanoid robots and connection to embodied intelligence remains largely unexplored. It remains unclear how to efficiently discover the optimal design of a generalist humanoid robot capable of performing a variety of tasks. More importantly, the necessity of addressing such co-design problems in the development of embodied humanoid robots has yet to be fully established.

This article provides a principled formulation of the humanoid co-design problem, emphasizing that **evolving physical structure is both feasible and essential for realizing embodied intelligence in humanoid robots**. Specifically, we formulate the humanoid co-design problem as a bi-level optimization. Such an optimizer can be integrated into the reasoning–acting architecture of an advanced controlling model, enabling an embodied humanoid robot to exhibit dexterity, mobility, perception, and intelligence.

Beyond the proposed formulation, we investigate an alternative perspective for realizing embodied humanoid robots based on predefined and manually specified designs. We analyze why such paradigms prevail in recent humanoid robotics research and examine the potential challenges of adopting co-design, particularly regarding algorithmic complexity, physical evaluation, and design scalability. To address these challenges, we introduce advancements in learning-based solvers, such as strategic robot structure exploration, the Sim2Real learning paradigm, and meta control policy, highlighting the feasibility of evolving humanoid robot architectures.

To understand the necessity of humanoid robot co-design, we investigate its unique advantages in facilitating robot morphology optimization, real-world task adaptation, and cross-disciplinary collaboration, examined from the perspectives of methodology, application, and community. To realize these key advantages, we identify open questions within the humanoid robot co-design problem, highlighting those that may be tractable with current methodologies in the short term, as well as those that may depend on long-term advances in emerging research domains.

## 2 Embodied Humanoid Robots

### 2.1 Architecture of Humanoid Robot

Humanoid robots are a specialized type of physical robot designed to replicate human-like functionality [21]. An ideal humanoid robot often has leggy designs in its *lower body*, featuring a *bi-pedal structure* that enables finishing locomotion tasks like walking, running, and maintaining balance. The *upper body* includes *dual arms* equipped with dexterous hands as end-effectors, allowing the robot to perform complex tasks that require precise manipulation and human-like hand movements. Their *sensor systems* commonly provide both *proprioception* and *exteroception*. Proprioception ensures internal body

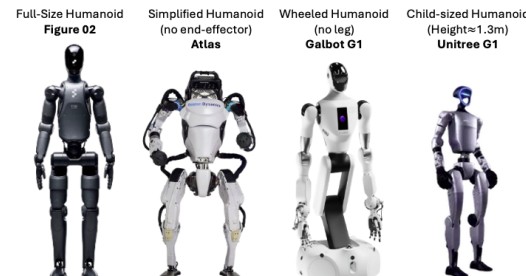

Figure 2: Examples of humanoid robots with varying physical structures are shown from left to right: full-sized, simplified, wheeled, and child-sized humanoid robots.

awareness by monitoring joint positions, angular velocity, and pose estimation, while exteroception enables perception of external states, such as LiDAR sweeps and RGB-D data.

While such designs are ideal, building and fine-tuning a humanoid robot's architecture typically demands substantial effort and resources. This often necessitates certain simplifications, such as 1) omitting end effectors in the dual arms, 2) constructing child-sized robots instead of full-sized ones, 3) utilizing a wheeled base for the lower body, and 4) downplaying the amounts and quality of sensors. Figure 2 provides illustrative examples of humanoid models.

**Desideratas for Humanoid Robot.** Given their human-like body structure, a fundamental requirement for humanoid robots is the ability to operate seamlessly within human environments. This enables them to collaborate closely with humans or take on dangerous or physically demanding tasks. During operation, the robot should exhibit natural behavior, adhering closely to human behavioral norms, even when performing long-term tasks across varying environments. These desiderata demand a control model with strong generalizability and adaptability, which traditional optimization-based control methods often struggle to achieve. Essentially, fulfilling this requirement calls for the development of embodied intelligence in humanoid robots.

## 2.2 Embodied Intelligence in Humanoid Robot

Unlike traditional approaches that rely on passively learning from fixed datasets, Embodied Artificial Intelligence (E-AI) requires agents to actively explore, interact with, and learn from their environment in a continuous and dynamic manner. Specifically, to enable the learning of an embodied humanoid robot, it often requires the robot to have four key abilities, including: 1) **Dexterity**: the ability to manipulate various objects with precision, delicacy, and intricacy. 2) **Mobility**: the capability to move and navigate through environments with different terrains and conditions. 3) **Perception**: the skill to gather, interpret, and understand environmental information from sensors. 4) **Intelligence**: the ability to process information, reason about sub-goals related to a given task, and adapt effectively to diverse tasks and environments.

These capabilities reflect not just the functionality of a humanoid robot in specific tasks like locomotion or manipulation but also emphasize generalization to a wide range of real-world scenarios, thereby advancing toward zero-shot deployment of humanoid robots for realistic applications.

## 2.3 Implementation for Embodied Humanoid Robot.

To achieve the above abilities, recent studies have implemented embodied humanoid robots using a two-layer architecture consisting of a high-level reasoning model and a low-level action model [22, 23, 24]. This hierarchical design is inspired by the functional organization of the human brain, where the cerebrum is responsible for logical reasoning and decision-making, while the cerebellum governs fine-grained motor control and coordination. These models are introduced as follows:

**Humanoid Robot Reasoning Model.** The reasoning model is typically implemented as a large-scale robotics foundation model designed to perform logical reasoning over the necessary steps for a robot to complete given tasks. The model takes as input natural language instructions describing the task, along with perceptual data, primarily visual signals collected from the environment. Its outputs consist of high-level plans to guide robotic execution. These may include sub-task descriptions and intermediate goals, as well as more detailed robotic outputs such as motion trajectories, grasp poses, and contact points (e.g., affordances). To learn this reasoning model, The training of the robotic reasoning model is typically achieved by fine-tuning a pre-existing Vision-Language Model (VLM) using refined robotic operation data, which is either collected through teleoperation or generated using synthetic data engines.

**Humanoid Robot Action Model.** The action model takes the outputs from the reasoning model (either as explicit data or implicit latent variables) as goals $g \in \mathcal{G}$ and predicts the corresponding control signals, such as joint angles or torques for the robot's joints at each time step. To learn action policies, previous methods commonly formulate the learning environment as a (partially observable) Markov Decision Process (MDP). $\mathcal{M} = (\mathcal{S}, \mathcal{A}, \mathcal{O}, P_{\mathcal{T}}, r, \mu_0, \gamma)$, where: 1) Within the state space $\mathcal{S}$, a state $s \in \mathcal{S}$ records the complete environmental information and the robot's internal states. 2) $\mathcal{A}$ denotes the action space, and action $a \in \mathcal{A}$ denotes the angles or torques at joints of the humanoid robot. 3) $o \in \mathcal{O}$ denotes the observations obtained from the robot's sensors, encompassing both proprioceptive inputs that reflect the humanoid's internal state and exteroceptive inputs that capture information about the external environment. 4) $r$ denotes the reward functions, which typically consist of penalty, regularization, and task rewards. In particular, the magnitude of the task reward

should closely reflect how well the robot accomplishes the given goal $g$. 5) $P_\mathcal{T} \in \Delta_{\mathcal{S} \times \mathcal{A}}^{\mathcal{S}}$ denotes the transition function as a mapping from state-action pairs to a distribution of future states. 6) $\mu_0 \in \Delta^{\mathcal{S}}$ denotes the initial state distribution. 7) $\gamma \in (0, 1]$ denotes the discounting factor.

Under this MDP, the humanoid action model can be represented as a meta policy $\pi \in \Delta_{\mathcal{S} \times \mathcal{G}}^{\mathcal{A}}$ that can scale to diverse goals $g\mathcal{G}$ under different environmental states $s \in \mathcal{S}$. During training, the goal is to maximize the expected cumulative discounted rewards:

$$\max_{\pi \in \Pi} \mathcal{J}(\pi, \mathcal{M}) = \max_{\pi \in \Pi} \mathbb{E}_{\mu_0, p_\mathcal{T}, \pi} \left[ \sum_{t=0}^{\infty} \gamma^t r(s_t, a_t, g) \right] \tag{1}$$

## 3 Position Proposal and Alternative Views

In the following sections, we introduce a co-design framework that jointly considers control policies and the evolution of humanoid morphology. Additionally, we present an alternative perspective: embodied intelligence should be grounded in predefined humanoid structures without evolution.

### 3.1 A New Perspective: Co-Designing Control and Evolution Policies

In natural environments, animals exhibit remarkable embodied intelligence, leveraging their evolved morphologies to learn and perform complex tasks [12]. Inspired by this, we argue that evolutionary principles should play an essential role in the development of embodied humanoid robots. While prior research commonly focused on perception, reasoning, and control within fixed robot structures, our position is that the robot's physical form itself should also be subject to optimization as a core component of its design. The simultaneous optimization of a humanoid robot's action model [1]) and physical components can be formulated as a co-design problem, integrating both control and morphology in the design process.

As illustrated in Figure 2, the robot co-design problem requires the joint optimization of both control policies and physical robot modules to maximize overall performance, while adhering to resource constraints such as cost [13]. When extending this framework to a learning-based setting, the co-design task typically involves a forward pass for training the control policy and a backward pass for updating the robot's physical parameters.

Specifically, during the forward process, we use the RL algorithm to optimize the goal-aware policy function by maximizing $\mathcal{J}(\pi, \mathcal{M}_\psi)$ in the objective (1), where the configuration of this learning environment (i.e., MDP) depends on a specific physical physical robot structure, denoted by $\psi \in \Psi$. Based on the policy performance, we conduct an inverse update on the robot's structure, which necessitates formulating humanoid robot co-design as a bi-level optimization problem. Moreover, the design often incorporates system-level constraints, which define strict requirements for the desired system behavior (e.g., resembling human behavior) or impose limitations on the resources (e.g., costs of robot modules) [13]. The optimization problem can be described as:

$$\max_{\psi \in \Psi} \max_{\pi \in \Pi} \mathcal{J}(\pi, \mathcal{M}_\psi) \text{ s.t. } f_c(\psi) \leq \epsilon \tag{2}$$

### 3.2 Alternative Views: Intelligence Arises from Fixed Humanoid Robots Structure

While the co-design approach provides a significantly broader design space for enhancing the performance of humanoid robots, most recent studies, spanning manipulation [25, 26, 27], locomotion [28, 29, 30, 31], and human motion imitation (i.e., teleoperation) [32, 8, 10, 11, 9, 33], continue to focus on controlling fixed, pre-defined humanoid platforms, without considering structural modifications to the robot itself. Even with the recent surge in exploring embodied intelligence in humanoid robots across a wide range of everyday tasks [34], *robotic co-design, as an effective technique in robotics research, remains largely underexplored* in this context [35]. This trend reflects a prevailing assumption that *"The predefined and fixed physical structures are sufficient for supporting the development embodied humanoid robots"*.

This perspective essentially treats the robot's structure as a fixed component of the environment's dynamics, which can be estimated and adapted to, but not actively optimized. Conceptually, this assumption is prevalent in the RL literature [36], which serves as a foundational algorithm for

---

[1]Since compare to robot acting, task reasoning is typically less dependent on the robot's detailed physical structure and updating a VLM is often costly, the reasoning model is generally not updated alongside changes to the robot's morphology.

learning-based control in humanoid robots and inherently shapes subsequent research directions. More importantly, in practice, there are significant challenges associated with co-designing humanoid robots, further reinforcing the reliance on manually designed, fixed-structure humanoid platforms.

**1) Complexity of the Co-design Problem.** In addition to learning control policy, the co-design problem incorporates an second-level optimization loop for refining the robot's physical design [13, 12, 37, 15, 17, 38]. For humanoid robots, this process becomes especially challenging due to their complex upper and lower body structures, which often involve varying configurations of motors, joints, sensors, and body components. Consequently, exploring the high-dimensional design space and identifying optimal configurations demands substantial computational resources. In many cases, due to the intricate interdependencies between the design and control parameters, the bi-level optimization in the co-design problem may struggle to converge. Without additional restrictions, the optimal solution may not be uniquely identifiable or even computationally tractable.

**2) Difficulties in Physical Evaluation.** To evaluate the optimality of a humanoid robot structure, it is crucial to deploy the robot in task-specific scenarios and assess how effectively it can adapt its control model to complete those tasks. This process often requires modifying certain components of the robot based on the proposed design. However, unlike simpler robotic systems, humanoid robots have highly complex and interdependent structures [39, 21, 40]. Modifying one part frequently leads to changes in the robot's overall physical configuration. For instance, adjusting the length of the thighs affects the robot's weight distribution and center of mass. These changes, in turn, influence both kinematic properties (e.g., motions and velocities) and dynamic characteristics (such as inertia and gravity models). The technical challenges in physical reconfiguration limit the feasibility of iterative structural design and evaluation in real-world applications.

**3) Limited Scalability Across Tasks.** Robotic co-design typically aims to enhance performance for specific tasks [13]. For instance, [17] optimized the leg length of a bipedal robot to achieve maximum walking velocity. Similarly, [15, 41] explored the joint optimization of mechanical structures and control policies to improve the locomotion capabilities of quadruped robots. However, embodied humanoid robots, with physical structures resembling those of humans, are designed to generalize across a wide range of tasks and environments within human workspaces. This requires multi-dimensional capabilities, including dexterity, mobility, perception, and intelligence (Section 2.2). The task-specific optimization frameworks commonly used in traditional robotic co-design cannot be directly applied to the inherently cross-task nature of humanoid robots. This lack of task scalability limits the overall utility of co-designed systems, particularly when targeting general-purpose humanoid platforms intended for reuse across diverse applications.

## 4 Feasibility of Co-Designing Humanoid Robot

To address the inherent challenges of humanoid robot co-design, this section investigates its feasibility by proposing a set of potential solutions. In particular, we introduce three key strategies: strategic exploration, the Sim2Real paradigm, and meta-policy learning. These approaches are aimed at tackling critical issues in co-design, including the complexity of joint design and control, the challenges of physical evaluation, and the limited scalability across diverse tasks. Moreover, by leveraging recent advances in control algorithms, simulated environments, foundation models, and decision-making policies, these strategies establish a robust foundation for the development of next-generation co-design algorithms for embodied humanoid systems.

### 4.1 Strategic Exploration under Constrained Design Space

In the co-design literature, genetic algorithms take a critical role in modifying robot structures via crossover, mutation, and replacement operations [12, 14, 15, 17, 38]. During this process, their underlying exploration mechanisms are inherently random. This randomness results in unstructured exploration, lacking informative priors or guidance toward designs that are more likely to yield higher-performing robots. This challenge becomes especially pronounced in unstructured and high-dimensional design spaces, such as those encountered in the development of humanoid robots, which induces computational burden (see alternative views in Section 3.2).

To address this limitation, *strategic exploration* has emerged as a promising approach for accelerating the search process. In the RL literature, a variety of algorithms have been developed to promote more efficient exploration [42]. For instance, recent work has proposed provably efficient strategies based on Bayesian updates [43, 44], the Upper Confidence Bound (UCB) [45], and uncertainty-

driven heuristics [46, 47], achieving significantly lower regret bounds compared to purely random exploration. Motivated by these advances, we propose formulating the physical physical robot structure process as an MDP, which enables the application of strategic exploration techniques to more effectively explore the design space of humanoid robots.

In addition to accelerating exploration, another important strategy for ensuring computational tractability and design identifiability is to *constrain the design space* with the following methods: 1) Rather than modifying the entire robot structure, *the exploration can be limited to a few key modules or components*. For instance, [18, 41] focused on optimizing the thigh and shank lengths of humanoid robots. Similarly, [37] investigated the configuration of various motors and the inertial parameters of robot links. Other studies, such as [48, 15], examined the parameters of parallel elastic knee joints. 2) From a theoretical perspective, a key approach to ensuring the convergence of bi-level optimization is to *bound the range of the design parameters*. By utilizing a compact design space and bounding the objective function in Equation 2, the Extreme Value Theorem (EVT) [49] guarantees the existence of a maximum. In this context, given a continuous objective function, the bi-level optimization can converge to an optimal (but not necessarily unique) solution $(\pi^*, \psi^*)$.

By strategically exploring a compact and bounded design space focused on key modules of the humanoid robot, the computational burden of humanoid robot co-design can be significantly reduced. This approach substantially alleviates the complexity of the bi-level optimization process (Section 3.2).

## 4.2 A Sim2Real Paradigm for Evaluation and Deployment

Guiding structural updates of humanoid robots typically requires evaluation signals from the current design. Traditional robot co-design studies often involve building physical hardware and assessing its performance in real-world tasks [20, 18, 15, 48]. However, as discussed in alternative perspectives (Section 3.2), this real-world design and evaluation approach is not directly applicable to humanoid robots due to the complexity of their physical structures.

Instead of relying on real-world evaluations, an alternative and effective approach is to conduct both the design and evaluation of robots within simulated environments. For example, studies such as [50, 51] and [15, 41, 17] utilize simulators like MuJoCo and Isaac Gym to learn control policies and evaluate the task performance of various physical robot structures. While simulation reduces the cost and complexity of building physical hardware, the Simulation-to-Reality (Sim2Real) gap remains a major challenge: a robot that performs well in simulation may not exhibit the same level of performance in the real world. Additionally, for humanoid robots with soft components, accurately simulating their morphology, especially for contact-rich interactions, remains difficult [52].

As a result, developing a feasible Sim2Real paradigm for humanoid co-design requires effective strategies to bridge the gap between simulation and reality. Key approaches include: 1) Extending *domain randomization and domain adaptation* techniques to humanoid control environments, which remains a critical direction for future research [53]. 2) *Developing simulation platforms with high photorealism and physical fidelity* to minimize the impact of the Sim2Real gap. For instance, recently developed simulation environments and engines such as RoboCasa [34], Isaac Lab [54], MuJoCo-Playground [55], Genesis [56], and ManiSkill [57] have demonstrated promising capabilities in accurately modeling real-world scenes and physical interactions. 3) Beyond explicitly modeling objects, scenarios, and physical laws, recent work has introduced the concept of the *World Function Model* [58, 59]. This paradigm treats simulation as a regression task, where the model predicts future states of the environment in response to perturbations (i.e., actions). This offers a more flexible and data-driven alternative to traditional simulators.

## 4.3 Controlling Humanoid Robot via Meta Policy

As discussed in Section 3.2, traditional approaches to robotic design and control have primarily focused on optimizing simple robots to complete specific tasks [13]. In contrast, embodied humanoid robots, as complex, legged, dual-arm systems equipped with numerous sensors, are designed to learn generalizable policies that can be applied across a wide range of tasks and environments. This fundamental difference renders previously proposed robot co-design methods not directly transferable to the problem of controlling embodied humanoid robots.

To enable the designed robot to perform effectively across multiple tasks, a critical approach is to learn a meta-policy $\pi$ that allows the humanoid robot to solve a wide range of everyday human tasks involving both manipulation and locomotion [34]. In the context of humanoid control, the meta-policy can be modeled as a goal-conditioned policy $\pi : \mathcal{S} \times \mathcal{G} \rightarrow \mathcal{A}$, where a goal $g \in \mathcal{G}$

may represent various forms of task specification, such as commands [31, 11, 29], target poses [32, 8], affordances [60, 61], or more generally, natural language descriptions of tasks [22, 24, 23]. Recent advances in generalizable decision models, such as decision transformers [62] and diffusion policies [1], provide effective backbone architectures for implementing such meta-policies, enabling flexible and scalable control across a wide range of goal specifications.

To integrate robotic co-design into policy learning, goals can be randomly sampled from a predefined goal pool during training. The robot's policy is then conditioned on each goal and adapted according to a proposed design configuration $\psi$. Similarly, during evaluation, the effectiveness of a given physical robot structure can be assessed by measuring how well it facilitates the learning of a policy that successfully achieves a diverse set of goals across varying environments.

## 5  Necessity of Co-Designing Humanoid Robot

In the previous section, we explored the feasibility of jointly modeling the robot's physical structure and its control policy, outlining key strategies that make such co-design tractable. In this section, we go a step further and argue for the necessity of co-design in the development of embodied humanoid robots from the perspective of metrology, application, and community.

### 5.1  Methodology: Principled Optimization of Robot Morphology

While significant progress has been achieved using pre-designed humanoid robots in both locomotion and manipulation tasks, there remains a lack of principled methods for evaluating the optimality of these designs. In practice, to determine robot morphology in Figure 2, *it commonly relies on engineers' intuition and experience, rather than through systematic optimization*. Such practice is often inefficient, since the robot's physical design is independent of the training of its planning models and control policies. The full capabilities and limitations of a given design often only become apparent when other research groups attempt to tackle more complex locomotion or manipulation tasks, revealing shortcomings that were not initially evident. These insights are used retrospectively to inform the development of robots, which typically progresses slowly due to the lack of systematic design methodologies. For example, in the initial design of the Unitree H1 robot, the limited degrees of freedom (DoFs) in its arms significantly constrained its ability to perform manipulation tasks that require rich interactions with objects. Recognizing this limitation, the developers addressed it in the subsequent version, Unitree H1-2 [63], by increasing the number of DoFs in each arm from 4 to 7. However, this design revision took over a year to implement.

In addition to improving the efficiency of robot development, co-design can significantly enhance its *efficacy*. From an algorithmic standpoint, allowing the robot's design $\psi$ to vary enables the learning algorithm to explore a larger joint search space over both morphology and control. This expanded space allows for the discovery of design-policy pairs that maximize overall performance:

$$\max_{\psi \in \Psi} \max_{\pi \in \Pi} \mathcal{J}(\pi, \mathcal{M}_\psi) \geq \max_{\pi \in \Pi} \mathcal{J}(\pi, \mathcal{M}_{\psi'}), \quad \forall \psi' \in \Psi \qquad (3)$$

Here, $\mathcal{J}$ denotes the expected return of policy $\pi$ in the environment defined by the morphology $\mathcal{M}_\psi$. This inequality highlights the potential performance gains from jointly optimizing both the robot's design and its control policy. Without principled optimization, manually identifying the optimal design $\psi^* \in \Psi$ is highly challenging, particularly for humanoid robots expected to perform diverse tasks and adapt to complex, real-world environments encountered in everyday scenarios. *To optimize both the efficacy and efficiency of discovering effective humanoid body designs, we argue that humanoid robot co-design is essential*.

### 5.2  Application: Adaptive Body Shaping for Real-World Tasks

In practice, the specific requirements for a humanoid robot's capabilities vary depending on the deployment environment of each real-world application. For example, in industrial settings, humanoid robots are often tasked with manipulating a variety of objects for relocation, rearrangement, or assembly. In such scenarios, dexterity becomes a critical factor, as robots must precisely and efficiently handle components of different shapes, sizes, and material properties [64, 30, 31, 65, 28]. In contrast, when deployed as patrol robots in environments such as university campuses or public facilities, humanoid robots must robustly navigate to different locations under diverse terrains (e.g., slopes, stairs) and environmental disturbances (e.g., dynamic obstacles or weather conditions). In these applications, robustness and mobility become the key performance criteria [66, 67, 26, 25]. Most importantly, *there exists a fundamental trade-off between dexterity and mobility* in the design of a humanoid robot's body structure. Achieving dexterous manipulation typically requires highly

flexible arms (with many DoFs) and delicate actuators. However, these design choices often result in increased weight and a higher center of mass, which can influence the robot's balance and reduce the efficacy of locomotion tasks.

A similar trade-off observed in humanoid robots can also be found in human physiology. For example, the body composition of boxers and runners differs significantly in terms of muscle distribution and weight allocation. These athletes often dedicate substantial time to optimizing their bodies to enhance the specific skills required in their respective sports. Just as athletes undergo intensive training camps to simultaneously develop both their physical form and technical skills, the co-design process of a humanoid robot's body and control policy can be viewed as a training camp for humanoid robots. By continuously adapting both morphology and behavior across diverse applications, we can dynamically tailor robot structures that are best suited for the target tasks and desired applications. *We argue that the co-design is essential for the humanoid robot to adapt specifically to its target applications.*

## 5.3 Community: Fostering Cross-Disciplinary Collaboration

Co-designing the control model and body structure of an embodied humanoid robot is fundamentally a multidisciplinary research topic. It encompasses: 1) *Machine learning expertise* for processing multi-modal sensory inputs, learning control policies, and performing high-level planning; 2) *Robotics design principles* for modeling and optimizing the robot's dynamics and kinematics; and 3) *Mechanical engineering knowledge* for the manufacturing of structural components and the integration of hardware systems. Each of these topics represents a significant research field with its own dedicated communities and research groups.

Traditionally, these research communities have evolved and been explored independently. For instance, machine learning is closely aligned with data-driven AI approaches, often emphasizing theoretical and methodological advancements in software systems. In contrast, robotics design and mechanical engineering emphasize the physical realization of hardware systems. However, when it comes to humanoid robot co-design, it fundamentally relies on the joint optimization of these domains, due to their deep and intricate interdependencies. Its advancement demands interdisciplinary collaboration, and the development of unified frameworks capable of optimizing control algorithms, morphological design, and hardware implementation in a coherent and efficient manner.

In recent years, cross-disciplinary collaboration, particularly under the AI+"X" paradigm, has played a vital role in advancing scientific and technological breakthroughs. For example, AlphaFold [68] exemplifies the synergy between deep neural networks and structural biology, revolutionizing protein structure prediction. OpenAI Five [69] and AlphaStar [70] integrate deep RL with the gaming and entertainment industry, pushing the boundaries of AI in complex, multi-agent environments. Similarly, Med-PaLM [71] bridges LLMs with medical knowledge, enabling AI-assisted healthcare solutions. These successes highlight the transformative potential of combining AI with domain-specific expertise. In this context, we argue that *humanoid robot co-design is essential for its unique capacity to foster cross-disciplinary collaboration across AI, robotics, and engineering.*

## 6 Open Questions in Humanoid Robot Co-Design

To encourage further exploration of humanoid robot co-design, we propose a set of open questions that may be addressed in both the short and long term.

**Open Questions in Short-Term.** We present open research questions that could be effectively tackled by leveraging emerging techniques and models.

1) *Efficient Representation for Robot Design.* Deriving concise and informative representations of data has been a key factor in the success of modern machine learning. In robotic learning tasks, recent studies have explored efficient representations for various types of multi-modal data, including language [72], 2D images [73], 3D shapes such as point clouds [74] and scenes [75, 76], and tactile information [77]. However, in most of these studies, the robot structure is fixed, and there has been limited exploration of efficient representations for robot morphology. This lack of focus significantly limits the efficiency of learning and adaptation in tasks involving adaptive robot design, particularly those that rely on updating deep neural networks.

2) *Benchmarking Robot Co-Design.* In robot co-design, Sim2Real training plays a crucial role by allowing the performance of co-design algorithms to be evaluated in simulated environments before deployment on physical hardware (Section 4.2). However, unlike other robotic manipulations and

location tasks with rich benchmarks [78, 79, 5, 80, 7], humanoid robot co-design, as a relatively new research area, lacks commonly applied benchmarks. Instead, prior studies often customize their tasks and environments according to specific goals, making it difficult to assess how well these algorithms generalize to other settings. While these case-specific studies provide valuable insights for physical robot structure, their applicability to other embodied humanoid robots, especially in diverse tasks and environments, remains uncertain. As a result, establishing a standardized benchmark for humanoid robot co-design emerges as an important and timely objective for the field, especially with the availability of simulation platforms such as Isaac Gym [81], MuJoCo [82], and Genesis [56].

3) *Design-Aware Policy Optimization.* In addition to learning meta-policies that can adapt to different tasks and environments, an important open question is how to develop design-aware policies $\pi_\psi$ : $\mathcal{S} \times \mathcal{G} \times \Psi \to \mathcal{A}$ [38]. Such policies are designed to generalize effectively across a range of different robot morphologies, enabling adaptive action control even when the physical structure $\psi$ changes. When a modification in the robot's body occurs, the policy can still perform reasonably well and, with minimal fine-tuning, adapt to the new structure. In this way, the policy can effectively serve as a morphology-aware controller, reducing the need for retraining from scratch whenever structural changes are introduced. This capability is crucial for co-design frameworks, where iterative updates to both control and morphology are expected. Ultimately, robust generalization across morphologies not only accelerates the Sim2Real transfer process but also enhances the practicality and scalability of humanoid robot deployment in dynamic, real-world environments.

**Open Questions in Long-Run.** We introduce promising robotic co-design research topics that depend on the advancement of other emerging areas, which are actively being studied but have yet to yield effective solutions.

1) *Co-Designs with World Models.* While learning-based co-design heavily relies on simulated environments, the simulators typically use manually specified semantics, rules, and physical laws, resulting in a non-negligible gap between simulation and the real world. To address this issue, recent studies [59, 58] have proposed building World Function Models (WFMs) that learn dynamics directly from real-world data. Inspired by the success of foundation models, WFMs are data-driven systems that automatically learn real-world physics and dynamics based on actions, without relying on human-designed assumptions. By jointly optimizing both the control policy and the robot morphology structure using WFMs, the gap between simulation and real-world application can be significantly reduced. However, developing reliable WFMs remains a challenging and long-term objective. As such, co-design based on WFMs is expected to be a major goal for future research.

2) *Co-Design Planning and Reasoning.* Current co-design studies primarily focus on the joint optimization of the action model and the robot structure. Although planning and reasoning models are integral components of embodied humanoid control systems, they are typically not subject to optimization during co-design and are instead used as pre-trained models. A major reason for this is the heavy computational burden associated with inferring and updating these large-scale Vision-Language Models (VLMs). In contrast, action models (i.e., policies) are relatively smaller, making their optimization more computationally manageable. As a result, a promising future direction is exploring the joint optimization of both reasoning and action models, so as the cover the entire process of. Achieving this will require the development of highly efficient inference and learning techniques (e.g., the use of Mixture-of-Experts (MoE) architectures), which remains an important and active area of ongoing research.

## 7 Conclusion

This paper advocates for a body-control co-design paradigm in humanoid robotics, emphasizing the joint optimization of both control strategies and physical morphology. Inspired by principles of biological evolution, we argue that co-design is essential for achieving embodied intelligence, enabling humanoid robots to adapt more effectively to diverse, dynamic real-world tasks. We demonstrate the feasibility of this approach through strategic exploration, Sim2Real transfer, and meta-policy learning, and highlight its necessity across methodological, application-driven, and interdisciplinary perspectives. By integrating co-design into the development pipeline, we can embrace the potential for more robust, generalizable, and intelligent humanoid systems. To guide future research, we outline key open questions, ranging from representation learning, benchmarking, and design-aware controlling to long-term integration with world models and reasoning systems. We position co-design as a foundational approach for developing intelligent, adaptable, and general-purpose humanoid robots capable of thriving in complex real-world environments.

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

# A   A Summary of Recent Studies in Robotic Co-Design

Table 1: The summary of recent studies and progress in robotic co-design.

| Designing Method | Robot Type | Designing Parameters |
|---|---|---|
| Meta RL [14] | Quadrupedal Robot | Thigh and shank lengths; gear ratios of the actuators. |
| Bayesian Optimization [15] | Quadrupedal Robot | Parameters of parallel elastic knee joint. |
| ADMM [48] | Quadrupedal Robot | Parameters of parallel elastic actuation (PEA). |
| Bayesian Optimization [41] | Quadrupedal Robot | Thigh and shank lengths. |
| Implicit Function Theorem [83] | Quadrupedal Robot | Link length; actuator poses. |
| Adjoint Method [84] | Quadruped and Hexapod Robot | Link lengths; actuator poses; robot's width and length |
| Evolution RL [17] | Lightweight Bipedal Robot | Thigh and shin lengths. |
| HZD Optimization [18] | Bipedal Robot | Thigh and shin lengths. |
| PPO [85] | Modular Soft Robot | 3D voxel-wise material assignments and spatial placement. |
| LLM-aided Evolution Search [86] | Modular Soft Robot | 3D voxel-wise material assignments and spatial placement. |
| Model Order Reduction [52] | Modular Soft Robot | The combination of actuator placement and pressure regulators. |
| DQN [20] | Modular Manipulating Robot | The combination of different modules. |
| RoboGAN [87] | Modular Locomoting Robot | Module type assignment on fixed-topology graph. |
| Graph Neural Network [88] | Modular Locomoting Robot | Size and position of limbs, type and range of joints |
| Particle Swarm Optimization [51] | Modular Locomoting Robot | Leg segment lengths. |
| PPO [89] | Modular Locomoting Robot | The combination of limbs. |
| Neural Graph Evolution [90] | Modular Locomoting Robot | The combination of different modules. |
| PPO [91] | Modular Locomoting Robot | Limb length and size; joint rotation range and torque limit. |
| Quadratic Programming [92] | Ariticulated Robot | Geometry and inertia of links; torque limits of joints. |
| PPO [50] | Legged Locomoting Robot | Link length and mass. |
| Genetic Algorithm [37] | ErgoCub2 Humanoid Robot | Motor types and link inertial parameters. |
| CMA-ES [93] | Freedom Endoskeletal Robot | Limb length; soft and rigid radii. |
| DGDM [94] | Sensor-less Jaw Manipulator | Manipulator finger geometry (represented as Bézier curves). |
| Binary Programming [13] | Autonomous Racing Drone | The combination of different modules. |

Table 1 summarizes recent studies in robotic co-design. We observe that most of these works focus on relatively simple robot types, such as modular and quadrupedal robots, with a limited number of design parameters considered. More importantly, the majority of these studies are tailored to specific tasks and environments. Extending these co-design methods to more complex humanoid robots operating across a diverse range of tasks and settings remains an important but largely unexplored challenge.

**No checklist is needed for position paper track (See Position Paper Track FAQ).**

