# OpenReview forum: "Embracing Evolution: A Call for Body-Control Co-Design in Embodied Humanoid Robot"
_NeurIPS.cc/2025/Position_Paper_Track — Submitted to NeurIPS 2025 Position Paper Track_

### Official Review · Reviewer_fzTG · 2025-07-26

**Significance:** 2
**Presentation:** 3
**Rating:** 4
**Confidence:** 4

**Summary:**

The paper argues that for humanoid robots, the physical hardware morphology of the robot needs and can be feasibly optimized (evolved) along with the control strategies for performing locomotion and manipulation with them, in order to develop embodied humanoid intelligence. The authors formulate the humanoid co-design problem as a two-level optimization problem, where the outer loop optimizes over designs and the inner loop learns control policies for a particular morphology. The paper then discusses the perceived challenges of co-design and proposes strategies for alleviating them.

**Strengths:**

- The paper is generally well written, and I appreciate that the background about humanoid robots and robot learning is provided in an approachable way, which could help readers from a variety of backgrounds engage with the paper.
- I believe that Section 3.2 indeed presents some of the most pressing technical and methodological challenges for co-design frameworks. Discussing and summarizing these challenges is a nice contribution.
- Additionally, the proposed solutions to the above challenges discussed in Section 4 appear to be technically reasonable, and although significant future research is required along each direction, pointing the community in such directions is valuable.

**Weaknesses:**

- One major area of potential improvement is the handling of the alternative position. The authors state that the alternative view is that “The predefined and fixed physical structures are sufficient for supporting the development [sic] embodied humanoid robots”. But, most of Section 3.2 focuses on a different alternative view, which can be summarized as “technical barriers prevent current practitioners from performing co-design for humanoid robots”. There is some discussion about the importance of co-designing humanoid robots in Section 5. But it’s confusing that the alternative view is first brought up in Section 3.2 and discussed later. Further, the points brought up in Section 5 do not address precisely the point in the alternative view.
- The novelty of co-evolving control and design is overstated. While this problem has not been well studied for humanoids, it also does not appear significantly different in problem formulation for humanoids.
- There are some awkward phrasings and imprecise notation in the paper, for example, “the RL algorithm” L159 could be phrased as “perform reinforcement learning” or omitted entirely (any generic way to optimize equation 1 is sufficient). Neither f_c nor epsilon are defined in Equation 2.

**Questions:**

- Do the authors have any thoughts or discussion about whether or not body-control co-design can enable or accelerate the progress of methodological or algorithmic development in embodied AI?
- I’m not sure that I agree that embodied AI requires agents to “actively explore, interact with, and learn from their environments in a continuous and dynamic manner”, as opposed to “passively learning from fixed datasets”. The majority of research still falls into the latter category and I believe this is an open question. Is this a critical distinction to make here? Why not allow the problem statement to encompass many possible approaches?

**Alternative Position:**

Yes, and alternative positions are well-considered and named but not addressed

**Author Identification:**

No.

**Context:**

3

**Discussion:**

2

**Ethics:**

["NO or VERY MINOR ethics concerns only"]

**Position:**

Yes, the paper argues for or against a position related to machine learning.

**Support:**

3

**Thoroughness:**

3

---

### Official Review · Reviewer_oa8r · 2025-08-09

**Significance:** 2
**Presentation:** 3
**Rating:** 6
**Confidence:** 4

**Summary:**

This paper argues that humanoid robots’ control strategies and physical structures should evolve together through a co-design process, rather than the alternative view of optimizing control while treating the physical structure as a fixed constant. It emphasizes the advantages of jointly optimizing both aspects, demonstrating the approach’s efficiency and effectiveness with concrete examples. The paper outlines key challenges in joint optimization and proposes potential solutions, while also highlighting open questions and future research directions prompted by co-design.

**Strengths:**

- The paper argues its position clearly, supporting it with concrete arguments and examples.
- The paper’s position is quite relevant to the community. It identifies the alternative position and points out that most of the current research follows it. This could inspire valuable discussions in the community.
- Major challenges of jointly optimizing for the control strategies and physical structure of humanoid robots are identified, and the paper presents potential solutions and research directions to address them.

**Weaknesses:**

- The necessity for jointly optimizing for control and the physical structure seems weak. While the arguments make co-design preferable, they do not show it is essential. For example, for the point adaptive body shaping, this can be addressed by having separate predefined designs for each specialized scenario as we do now, not necessarily co-designing control and morphology. Regarding the point on fostering cross-disciplinary collaboration: why is such collaboration valuable beyond serving the purpose of co-designing control and morphology? The argument feels circular—co-design is said to be necessary because it fosters collaboration, and collaboration is deemed important because it supports co-design.

- The impact of jointly optimizing control and physical structure in existing work is not well analyzed. Is this co-design mechanism necessary for robots beyond humanoids? If so, why is it necessary? If not, what makes humanoid robots uniquely suited to require such co-design?

**Questions:**

- Is co-design necessary only for humanoid robots, or for robots in general? For existing work on co-design for other robot morphologies like quadruped robots, why and how is it necessary?
- For hardware developers, avoiding overfitting the physical structure to a narrow set of tasks would require optimizing control strategies across a diverse task set following the proposal in Section 4.3. Is this realistically feasible for companies primarily focused on hardware?
- The paper proposes learning a meta-policy across diverse tasks to prevent overfitting the co-design to limited scenarios in Section 4.3. However, wouldn’t this make it harder for the joint optimization to converge—an issue the paper identifies as a key challenge in Section 3.2.1?

**Alternative Position:**

Yes, and alternative positions are well-considered and addressed by the argument

**Author Identification:**

No.

**Context:**

3

**Discussion:**

3

**Ethics:**

["NO or VERY MINOR ethics concerns only"]

**Position:**

Yes, the paper argues for or against a position related to machine learning.

**Support:**

3

**Thoroughness:**

4

---

### Official Review · Reviewer_abRD · 2025-08-29

**Significance:** 3
**Presentation:** 3
**Rating:** 8
**Confidence:** 2

**Summary:**

This position paper argues that humanoid robots should be co-designed—optimizing control policies and physical morphology together—rather than developing control systems for fixed robot structures.

Core Position: The authors argue that evolving the physical structure alongside control strategies is essential for achieving true embodied intelligence in humanoid robots, drawing on biological evolution, in which organisms adapt both form and behavior to their environments.

Main Contributions:

1. Formulation: Presents humanoid co-design as a bilevel optimization problem where control-policy optimization feeds into morphology optimization, subject to resource and behavioral constraints.

2. Feasibility Analysis: Addresses three major challenges that have limited co-design adoption:
(1) Computational complexity, (2) Physical evaluation difficulties, (3) Limited task scalability.

3. Necessity Arguments: Makes the case from three perspectives:
(1) Methodological, (2) Application, (3) Community.

**Strengths:**

The paper presents a compelling argument advocating for the integration of body-control co-design in humanoid robots, emphasizing its importance for advancing embodied intelligence.
The authors clearly argue for the necessity of simultaneously optimizing control strategies and robot morphology, drawing inspiration from the process of biological evolution. This perspective is well-supported by solid reasoning and provides effective approaches to achieve co-design. By organically combining control and morphology design, the authors suggest that humanoid robots can better adapt to real-world tasks and dynamic environments.
The argument is based on rigorous logical reasoning, and the paper also proposes practical methodologies such as strategic exploration, Sim2Real transfer, and meta-policy learning to ensure the practical feasibility of the co-design approach.

**Weaknesses:**

1. Implementation Challenges: The paper could provide more concrete examples of how co-design has been successfully implemented in current humanoid robotics or other fields. It would help readers visualize the practicalities of the proposed approach.
2. Alternative Approaches: The paper mainly focuses on co-designing both control and morphology. An alternative view that could be discussed further is the possibility of leveraging modular, adaptive control models that do not require redesigning the physical structure but instead adapt to varying tasks and environments using more flexible, robust control algorithms.
3. Scalability and Generalization: Although the co-design framework promises improved performance, scalability across diverse tasks and environments is a significant challenge. The paper could address how co-design might be generalized to a broader range of robots or non-humanoid systems, as the primary focus here is humanoid robots.

**Questions:**

Feasibility of Real-Time Co-Design: Given the significant computational complexity involved in co-designing both the robot's control and morphology, how do the authors foresee scaling these techniques for real-time applications, especially in dynamic environments with changing task requirements?

**Alternative Position:**

Yes, and alternative positions are trivial straw-man arguments

**Author Identification:**

No.

**Context:**

3

**Discussion:**

4

**Ethics:**

["NO or VERY MINOR ethics concerns only"]

**Position:**

Yes, the paper argues for or against a position related to machine learning.

**Support:**

3

**Thoroughness:**

3

---

### Note · Authors · 2025-08-22

**1-10 Additional Comments:**

The authors are indeed proposing an important proposal, requiring cross-disciplinary collaboration that spans both hardware and software design in AI. However, this type of research may feel unfamiliar to much of the ML community. The interdisciplinary nature of the proposal could lead to misunderstandings or undervaluation, despite its potential to open up important new directions for future research.

**1-11 Submit Again:**

Definitely yes

**1-1 Submission Process:**

4

**1-2 Next Year:**

Yes, the authors would definitely be interested in the position paper track. In terms of scientific discovery, proposing new questions and exploring potential future research directions is critically important, even if not everyone agrees with the proposals. In fact, sparking discussion and shaping future inquiry can be just as valuable, if not more so, than solely focusing on solving existing problems.

**1-3 Future Development:**

I have the following ideas that might be helpful:
1. The authors believe a more engaged discussion between the authors and the reviewers will be helpful.
2. The authors believe that evaluating papers in the position paper track will present a significant challenge. Relying solely on a scoring system to assess quality could be problematic. While such systems may work well for main track papers, where substantial experiments and novel theories are typically expected, they may not be suitable here. Without empirical validation or concrete results, it becomes much more difficult to make objective judgments. As a result, the roles of ACs and SACs will be even more critical in this track, requiring careful interpretation and discussion to ensure fair and thoughtful evaluation.

**1-4 Interest:**

["Panel discussions with other position paper authors", "Structured debates on controversial topics", "Workshops for developing position papers"]

**1-5 Thoughtful:**

6

**1-6 Supportive:**

7

**1-7 Technical Aspects Versus Position:**

8

**1-8 Gate Keeping:**

8

**1-9 Camera Ready Changes:**

1. Clarifying the Alternative Viewpoint:
We will clarify that Section 3.2 consists of two deliberate parts. The first introduces the alternative perspective that fixed physical structures are sufficient for building embodied humanoid robots, supported by existing studies. The second part dives deeper into the underlying technical barriers that explain why practitioners often rely on fixed structures. This structure is important because it sets up the motivation for Sections 4.1 to 4.3, which present solutions to these barriers. Section 5 then builds on this by arguing that co-design is not just feasible but also essential.

2. Differentiating from Traditional Co-Design Approaches:
We will discuss how our study differs significantly from traditional co-design literature. Their approach is framed within the context of modern embodied AI, focusing on AI-driven methodologies. Unlike conventional methods relying on human intuition and manual tuning, their pipeline is data-driven and algorithmic, better suited for the complexities of humanoid robots.

3. Clarifying the Role of Active and Passive Learning:
We will clarify that while most current research still relies on passive learning, embodied AI typically requires active interaction with the environment, often through reinforcement learning. They emphasized that the paper does not argue for the superiority of active learning, but rather reflects the requirements of training embodied humanoid agents.

4. Justifying the Necessity of Co-Design:
The authors will reaffirm that co-design is essential for developing generalist humanoid robots capable of handling diverse tasks and environments. While predefined designs may work in specialized scenarios, they are often suboptimal and based on non-optimized human intuition. Co-design, in contrast, enables the explicit optimization of both morphology and control, thereby improving performance and adaptability.

5. Provide concrete examples of implementing co-designs.

**3-1 Review Response1:**

Reviewer fzTG

**3-2 Reaction To Review1:**

We believe the reviewer's concerns largely stem from a misunderstanding, and most of these points are already addressed in the paper:

1 *One major area of potential improvement is the handling of the alternative position. ..But, most of Section 3.2 focuses on a different alternative view ...*

As noted in the review, the first half of Section 3.2 clearly states that "the predefined and fixed physical structures are sufficient for supporting the development of embodied humanoid robots,". Building on this observation, the second half of Section 3.2 seeks to **delve deeper into the phenomenon** and identify the key underlying factors. **It is important to emphasize that presenting these technical challenges is essential**, as Sections 4.1, 4.2, and 4.3 correspond to our proposed solutions for overcoming these barriers**.

2 *..While this problem has not been well studied for humanoids, it also does not appear significantly different in problem formulation for humanoids.*

In these articles, we emphasize advancing co-evolving and co-design methodologies for humanoid robots within the **context of modern embodied AI**. This methodological foundation significantly differs from that of traditional co-design literature (see lines between 40 to 42). For instance, conventional approaches often involve manual design and evaluation processes that heavily rely on real-world heuristics and human experience.

3 *Do the authors have any thoughts or discussion about whether or not body-control co-design can enable or accelerate the progress of methodological or algorithmic development in embodied AI?*

Yes, in Section 5.3, specifically between lines 362 and 378, we discuss the methodological and algorithmic developments from the perspective of multidisciplinary research that bridges machine learning, robotics, and mechanical engineering. In this context, body-control co-design presents a unique opportunity to accelerate progress in embodied AI.

**3-3 Review Response2:**

Reviewer oa8r

**3-4 Reaction To Review2:**

Similarly, we believe most of this reviewer's concerns have already been addressed in the paper.
1. *The necessity for jointly optimizing for control and the physical structure seems weak...for the point adaptive body shaping, this can be addressed by having separate predefined designs for each specialized scenario. *

While we have separate designs tailored for different environments, we would like to clarify that the ultimate goal of developing modern embodied humanoid robots is to create generalist agents such that robots are capable of performing a variety of tasks across diverse environments (lines 99 to 104.. Predefined designs are largely based on human intuition and domain knowledge. However, there is no guarantee that they are optimal for specific scenarios.

2. *... why is such collaboration valuable beyond serving the purpose of co-designing control and morphology?...*

 We believe the argument holds true only in the forward direction: co-design is essential because it fosters cross-disciplinary collaboration. Indeed, many of today's breakthroughs stem from such "AI + X" collaborations (see lines 370–378). However, collaboration can take many forms beyond co-design (see lines 370–378). These collaborations are valuable in their own right and do not solely depend on supporting co-design.

3. *...Is this co-design mechanism necessary for robots beyond humanoids? If so, why is it necessary? If not, what makes humanoid robots uniquely suited to require such co-design?*

Yes, Co-design mechanisms are indeed necessary for other types of robots (see references, lines 40–41). What makes humanoid robots unique is that 1) they introduce new challenges (lines 183 to 212) to the co-design process, making previous methods less effective.
2) They are capable of operating seamlessly within human environments and handling everyday tasks as generalist robotic agents, abilities that many other robots lack (see lines 28–29 and 87–94).

**3-5 Review Response3:**

Reviewer abRD

**3-6 Reaction To Review3:**

We thank the reviewers for their thoughtful feedback. We address each point below and clarify how the concerns are already discussed in the paper or can be further emphasized in a revision.

1. *... more concrete examples of how co-design has been successfully implemented in current humanoid robotics or other fields.*

In Section 2.1 (lines 82–86) and Table 1 (Appendix A), we provide concrete examples of simplified humanoid prototypes and summarize co-design across quadrupeds, bipedal robots, soft robots, and modular robots. While humanoid-specific examples remain limited due to the nascent nature of this direction, we highlight that humanoids are the next frontier where co-design has not yet been systematically explored.

2. *... An alternative view that could be discussed further is modular, adaptive control models without redesigning the body.*

This is precisely the role of Section 3.2, “Alternative Views.” As described in lines 167–176, we explicitly acknowledge the prevailing assumption that “predefined and fixed physical structures are sufficient”. We then detail the rationale for this perspective, including complexity, physical evaluation, and scalability challenges (lines 181–212).

3. * The paper could address how co-design might be generalized beyond humanoids.*

We appreciate this suggestion. Our discussion of scalability challenges (Section 3.2, lines 207–212) already highlights that task-specific optimization frameworks used in quadrupeds or modular robots are not directly transferable to humanoids.

4. * ... How do the authors foresee scaling these techniques for real-time applications?*

We agree that real-time feasibility is critical. While our paper does not claim that full real-time co-design is currently practical, Section 4.1 (strategic exploration) and Section 4.2 (Sim2Real) already propose concrete steps toward tractability. In future work (Section 6.1–6.2), we envision integrating world models [58, 59] to accelerate online adaptation.

---

### Meta-Review · Area_Chair_pW3T · 2025-09-10

**Rating:** 6
**Confidence:** 4

**Strengths:**

The reviewers considered the paper compelling, well-supported, relevant, and a contribution to the field

**Weaknesses:**

Reviewers expressed concern that the paper needs a stronger discussion of alternative positions, needs to clarify the extent to which the authors feel co-design is essential, and should provide more concrete examples of past co-design efforts.

**Questions:**

Some question themes include:

Why do the authors focus on humanoids and would this argument equally hold in other domains?

and

To what extent is largescale & robust co-design feasible for hardware producers?

**Thoroughness:**

3

---

### Decision · Program_Chairs · 2025-09-26

Reject